# Development and Rasch Analysis of the 18-Item Health Resilience Profile (HRP)

**DOI:** 10.3390/nu15040807

**Published:** 2023-02-04

**Authors:** Natalie M. Papini, Myungjin Jung, Minsoo Kang, Nanette V. Lopez, Stephen D. Herrmann

**Affiliations:** 1Department of Health Sciences, Northern Arizona University, Flagstaff, AZ 86011, USA; 2Department of Health, Exercise Science, and Recreation Management, The University of Mississippi, Oxford, MS 38677, USA; 3Department of Internal Medicine, University of Kansas Medical Center, Kansas, KS 66160, USA

**Keywords:** psychometric evaluation, Rasch analysis, resilience, health coaching, scale development

## Abstract

Existing resilience measures have psychometric shortcomings, and there is no current gold-standard resilience measure. Previous work indicates adults enrolled in a health coaching program may benefit from a resilience measure that is tailored and contextualized to this sample. This two-part study aimed to develop and evaluate a resilience instrument focused on health behavior change in adults in a health coaching program. Two studies were conducted to (1) create a resilience instrument (Health Resilience Profile; HRP) specific to adults attempting health behavior change (*n* = 427; female = 83.8%; age = 44.5 ± 11.9 years) and to (2) optimize the instrument performance using Rasch analysis (*n* = 493; female = 62.1%; age = 49.8 ± 12.5 years). Study 1 identified two issues: (1) four unacceptable misfit items and (2) inappropriate rating scale functioning. Study 2 evaluated an improved instrument based on the outcome of study 1 resulting in one more misfit item, and unidimensionality was supported. The new four-category rating scale functioned well. The item-person map indicated that item difficulty distribution was well matched to participants’ resilience level, and items were free from measurement error. Finally, items did not show differential item functioning across age, sex, alcohol use, and obesity status. The 18-item HRP is optimized for adults in a health coaching program.

## 1. Introduction

The optimal dose of health coaching for physical activity, dietary, stress management, and smoking behavior change is 30 or more sessions over the course of 6 to 12 months [1]. However, attrition is common in lifestyle modification programs focused on physical activity and dietary habits [2,3]. Attrition or “dropout” can be identified when an individual refuses or fails to return to treatment or is removed for lack of cooperation [4]. People enrolled in lifestyle management program report a variety of social, demographic, and behavioral barriers for participation and reasons for drop out [5]. Given that the reported attrition estimates for both mobile health “mHealth” and in-person delivered treatments are between 20% and 70%, it is essential for health coaching programs to adequately identify, measure, and cultivate protective factors that minimize rates of attrition [5,6].

One such protective factor could be resilience, defined as an individual’s ability to manage and adapt to significant sources of stress or the ability to “bounce back” after significant barriers or adversity [7,8,9]. Resilience is a crucial component of positive psychology coaching, which Green and Palmer (2014) define as an “evidence-based coaching practice informed by the theories and research of positive psychology for the enhancement of resilience, achievement, and well-being” [10].

Different from goal-related self-efficacy (the extent to which an individual believes a goal can be attained), in the context of health coaching, resilience focuses on an individual’s ability to persevere through challenges and adversity as it occurs concerning the health goal(s) [11]. Resilience is also different from grit, defined as perseverance and passion toward long-term goals including consistency of effort [12,13,14].

Less attention has focused on cultivating resilience among individuals participating in health coaching programs that target diet, physical activity, and other lifestyle changes. Findings indicate that coaching can support increased resilience in various populations and contexts, such as executive leaders, employees enrolled in an employee wellness program, and middle school children [15,16,17,18]. However, findings from resilience interventions should be interpreted with caution since there no “gold standard” resilience measures [9]. There are a wide range of known barriers reported by individuals in health coaching programs that address diet and physical activity including navigating social settings while staying committed to health goals, lack of knowledge on physical activity behaviors, and monotony and boredom with dietary behavior change [19]. Increasing resilience may be a worthwhile pursuit for health coaching programs to help individuals overcome and persist in light of these barriers.

The first step toward integrating resilience into health coaching programs is to ensure that tools adequately measure resilience specific to the target population. The development and evaluation of a resilience instrument can employ a Rasch model, a more advanced, probabilistic measurement model [20,21]. Rasch analysis permits the examination of the response categories’ functioning, the measure’s unidimensionality, and the measure’s targeting [22]. The Rasch model is preferred over traditional methods because it accounts for the difficulty level of individual items and transforms participant responses based on ordinal scales into interval scales via logits [21]. Additionally, the Rasch model shows individual item difficulty, spread, redundancy, and gaps across a wide range of person-ability scores through a visual representation in an item-person map [23].

Several instruments exist to measure resilience in a variety of populations; however, Windle and colleagues (2011) noted that all measures they reviewed required psychometric improvements and further validation work [9]. Of the several resilience options, the Connor–Davidson Resilience Scale (CD-RISC) [24] and the Resilience Scale for Adults ranked highest [25]. The psychometric properties of the 25-item CD-RISC were previously evaluated using Rasch modeling within adults enrolled in a health coaching program, and a ceiling effect (i.e., was not effective at measuring moderate to high levels of resilience) was observed [26]. This result aligned with previous research utilizing the CD-RISC-25, which observed a ceiling effect in a sample of nonclinical adults in Spain [27]. In health coaching populations, the lack of a resilience measure specific to health behavior change and health-related goals could limit accurate baseline assessment and result in a reduced ability to measure resilience change over time as a person progresses through a health coaching program. Papini and colleagues (2020) noted the need for future research to develop and test an improved inventory that measures a broader range of resilient behaviors within adults attempting behavior change through the support of a health coaching program [26]. Therefore, the purpose of the current two-part study was to develop and evaluate a resilience measure for adults enrolled in a health coaching program. Part of the development of this measure was to adjust the phrasing of resilience items to reflect scenarios and situations that would require resilience in a health coaching program. The first study aimed to develop and evaluate the initial 23-item Health Resilience Profile (HRP). Results from study 1 were used to produce the final 18-item HRP instrument. The purpose of study 2 was to assess the psychometric properties of the 18-item HRP using Rasch analysis.

## 2. Materials and Methods

Development and validation of the HRP occurred in two phases. The first study included the development of items contextualized to adults in a health coaching program and the evaluation of those items. The aim of study 1 was to assess the measurement properties of this new instrument.

The new HRP was created based on previous findings [26,27] to retain the most difficult item (i.e., “I am not easily discouraged by failure”) and a Delphi method (structured group communication processes aimed at gathering expert feedback to come to a consensus on instrument content, wording, and application) to gather expert feedback and consensus from six subject matter experts [28]. A total of 23 HRP items were used and evaluated in study 1. Study 2 aimed to fix limitations observed in the HRP from study 1 and then reexamine the measurement properties using Rasch modeling. Studies 1 and 2 were approved by the Institutional Review Board (IRB) at Northern Arizona University. These studies were not preregistered. Data and study materials are available upon request.

### 2.1. Rasch Calibration Data Analysis

Both studies followed a similar data analysis methodology. First, descriptive statistics were performed using SPSS (version 27; SPSS Inc., Chicago, IL, USA). A two-facet Rasch rating scale model was used to evaluate the HRP with the Winsteps for Windows program (version 3.65). The second model-data fit was evaluated by examining Infit and Outfit statistics [29]. Items with Infit and Outfit values less than 0.5 (i.e., too little variation in responses) or greater than 1.5 (i.e., inconsistent responses) were considered a poor fit [29]. Third, a unidimensionality of the scale and local independence of the item was also evaluated based on Linacre’s guidelines [30]. The dimensionality of the scale and local independence of the item was examined by conducting Rasch factor analysis using Principal Component Analysis of the standardized residuals and a residual correlation, respectively. The unidimensionality is satisfied if the first contrast (component) is not much bigger than two eigenvalues. The items in the scale are locally independent of each other if a residual correlation is not greater than 0.7. Fourth, the function of the rating scale was analyzed to determine whether the existing instrument response category was appropriate. The evaluation criteria were as follows: (1) Was there regular observation distribution such as unimodal, bimodal, or slightly skewed distribution? (2) Did the average logit score measured for each category increase as the category increased? (3) Was the Outfit mean square residual appropriate for each category (Outfit statistics < 2.0)? (4) Were the category thresholds (i.e., boundaries between rating categories) ordered? [31]. Fifth, an item-person map distribution was examined. The map visually illustrates logit scores (transformed raw scores that permit more precision in analysis since it captures the relationship between the item responses and the participant’s ability level) of resilience item difficulty relative to a person’s resilience level on the same scale, thus allowing the comparison of these measures. Sixth, each item’s difficulty was calculated during the calibration process in logits. The higher the logit score, the more difficult the participants perceived the resilience item. Item separation index and item separation reliability were also examined to evaluate how well items were separated along the measurement scale (separation score > 2.0 desired) and the plausibility of replicating item placements in another sample (reliability scores > 0.8 desired). Seventh, the person’s level of resilience was estimated, and the higher logit score indicated a higher level of resilience. Person separation index (the degree to which individuals within a sample can be distinguished from one another on the basis of their responses; separation score > 2.0 desired) and person separation reliability (measures the reliability of the person separation index, and indicates how reliable it is in distinguishing between individuals; reliability scores > 0.8 desired) were also investigated to determine the quality of the HRP as they both indicate how well it is able to distinguish between participants. Eighth, differential item functioning (DIF) analysis was conducted to demonstrate that items in HRP function differently by variables such as age (young adults (18–35 years)/middle-aged adults (36–55 years)/older adults (≥56 years)), sex (male/female), alcohol use (yes/no), and obesity status (person with obesity/person without obesity). Items were considered biased when they exhibited both substantive (i.e., Mantel–Haenszel (M-H) DIF size > 0.64 logits) and statistical significance (*p* < 0.001). If the M-H DIF is larger than 0.64 logits, the functions were different among the groups. Lastly, convergent and known-group difference validity evidence was examined for study 2.

### 2.2. Study 1: Instrument Development and Evaluation

#### Participant Recruitment and Procedure

Participants (*n* = 427) were adults in a health coaching program focused on nutrition, physical activity, and lifestyle change. See Table 1 for demographic information of participants who completed study 1. Participants were recruited by email or through a private Facebook group affiliated with the health coaching program. Approximately 12% of the sample had missing data for some demographic items but completed the resilience scale and were included in the analysis.

### 2.3. Measures

Participants were instructed to complete an online survey hosted using Qualtrics which contained basic demographic questions, the 23 HRP items, a short (single-response) physical activity assessment, and the 10-item Perceived Stress Scale (PSS) [32]. The 10-item PSS was used to assess the convergent validity evidence of the HRP, and the single-response Physical Activity Questionnaire was used to examine a known-group difference validity evidence [33]. This single-response Physical Activity Questionnaire invites participants to select one of eight possible responses that best matches their current level of physical activity. Participants who respond with the fifth response or higher are meeting or exceed physical activity guideline recommendations [33].

Demographics. Participants were asked to complete demographic questions, including age, sex, alcohol use, weight, and height. For demographic information on study samples, see Table 1.

Perceived Stress Scale (PSS) [32]. The 10-item PSS was used to measure participant perceptions of stress. The PSS has demonstrated strong validity and reliability across a wide range of participant samples [34,35]. Items on the PSS are rated on a 5-point scale ranging from “never” (0) to “almost always” (4). PSS-10 scores range from 0 to 40, and scores of 20 or greater are considered higher stress indicative of therapeutic help [36,37].

Physical Activity [33]. A single item was used to measure participants’ physical activity levels. Participants were instructed to select the statement that best described their physical activity level. Definitions with examples of “moderate” and “vigorous” activity were provided to participants. Physical activity levels were divided into two groups: Low vs. Moderate-Vigorous.

Health Resilience Profile (HRP-23). In study 1, the HRP was a 23-item measure of resilience around health goal attainment and health resilience. HRP items were developed based on previous research that used Rasch analysis to assess the psychometric properties of the CD-RISC in a sample of adults enrolled in a health coaching program [26]. The study team removed items that loaded on the same difficulty level and worked to contextualize the remaining items to be more specific to goal attainment for adults receiving health coaching in nutrition, physical activity, and lifestyle behavior change. Additional items were added to the HRP-23 that were consistent with resilience, defined as a person’s ability to “bounce back” in the face of adversity and specific to adversity clients experience when trying to reach a health goal. The original tool included a 5-point Likert scale that asked participants to rate items from “strongly disagree” (1) to “strongly agree” (5). Scores on the original HRP used in study 1 ranged from 23 to 115, where lower scores indicated lower health resilience and higher scores indicated higher health resilience. Findings from study 1 informed changes made to the HRP in study 2.

### 2.4. Profile by Sanford Health Coaching Program

Participants in this study were enrolled in the Profile by Sanford health coaching program. This personalized health coaching service helps individuals reach health goals through one-on-one meetings that target nutrition, physical activity, and lifestyle behaviors (such as stress management). Individuals in the program are encouraged to attend a 30-min weekly coaching appointment throughout a 1-year membership and can meet with a single health coach or use different health coaches. Health coaching appointments include reviewing past conversations, topics, and goals from the previous week, discussing new education and behavior topics, modifying existing goals, or setting new goals for the next week. The Profile by Sanford program is efficacious in reducing weight in endometrial cancer survivors and improving fruit/vegetable consumption, home food preparation, and life satisfaction in employees enrolled in a worksite wellness program [38,39].

## 3. Results

### 3.1. Model Data Fit

In the initial analysis, two items were flagged due to high infit and outfit statistics, meaning the items did not perform well. Item 16 had an infit value of 1.90 and an outfit value of 2.08, and item 1 had an infit value of 1.65 and an outfit value of 1.90. When a second Rasch analysis was performed without items 16 and 1, item 8 registered an even larger infit value of 1.60 and an outfit value of 1.74. When a third Rasch analysis was performed without items 16, 1, and 8, item 18 registered an even larger infit value of 1.53. Therefore, items 16, 1, 8, and 18 were removed. The final analysis with 19 items showed individual item infit statistics ranging from 0.72 to 1.33, which fall within the acceptable range of 0.5 to 1.5. The individual item outfit statistics ranged from 0.73 to 1.29, also indicating appropriate fit. In addition, the items of this scale were locally independent of each other (<0.7) and the unidimensionality of this scale was satisfied (an eigenvalue on the first contrast was 2.4), indicating that the items measure a single construct or trait (e.g., resilience).

### 3.2. Rating Scale Functioning

The function of the rating scale (i.e., Likert response format) was analyzed to determine whether the five response categories were appropriate for the items. Regular observation distribution was found, the average logit measures advanced as the category increased, and the outfit statistics fell within the desired range of <2.0. However, the category thresholds (i.e., boundaries between rating categories) were not arranged in sequence (see response ‘3’ in Panel ‘A’ of Figure 1) indicating that the 5-category rating scale did not function well (Table 2) and should be improved.

### 3.3. Item-Person Map

Panel ‘A’ of Figure 2 depicts the item-person map for study 1. The item-person map is a visual representation of the findings for both person ability (left side) and individual item difficulty (right side) on the same continuum using logits. The logit score for item difficulty level is shown on the right side of the map, indicated by question number. The logit scale on the left side of the map, indicated by “#” and “.” symbols (each ‘#’ is 4, each ‘.’ is 1), displays the resilience level of persons.

### 3.4. Item Difficulty and Person Ability

The resilience item difficulty estimates ranged from −3.18 to 3.15 logits. A higher logit score indicated a more difficult resilient item. The most difficult resilience item was question 22 (“I get discouraged when I’m not making progress toward my health goal(s)”) with a 3.15 logit (SE = 0.08). The least difficult resilient item was question 15 (“I feel a sense of pride when I reach a health goal”) with a −3.18 logit (SE = 0.11) (see Table 3). The item separation index was 18.27, indicating that the HRP items were well distributed across the measurement scale. The separation reliability was 1.00, indicating a high degree of confidence in replicating the position of the items within measurement error for another sample.

The individual level of resilience was estimated by logit, where a higher logit represented a higher resilience level. The average resilience level was 1.11 (SD = 1.24). The individual level of the resilience estimates ranged from −2.68 to 8.58 logits. Person separation was 2.80, denoting that person’s abilities are well separated along the measurement continuum. Person separation reliability was 0.89, signifying an acceptable degree of confidence in replicating the placement of persons within a measurement error.

### 3.5. Differential Item Functioning (DIF)

According to the DIF analysis, no items function differently in any subgroups (i.e., by age group, alcohol use, and obesity status). This indicates that the items function properly regardless of the subgroups. In other words, the items did not show bias based on different individual characteristics and demographics.

### 3.6. HRP Revisions Following Study 1

After reviewing infit/outfit statistics, rating scale functioning, and item difficulty level, several revisions were applied to the original 23-item HRP. Four items (1, 8, 16, 18) were removed from the instrument. Rating scale functioning from study 1 indicated that a 5-point Likert scale did not function appropriately and that a 4-point scale would be optimal. As such, the 5-point Likert scale (strongly disagree, disagree, neither agree nor disagree, agree, strongly agree) was revised to a 4-point scale upon removal of the “neither agree nor disagree” response option. A detailed summary of revisions for each item is provided in Table 4.

### 3.7. Study 2: Rasch Calibration of the HRP

#### 3.7.1. Participant Recruitment and Procedure

A total of 493 adults were recruited from a health coaching program focused on nutrition, physical activity, and lifestyle change (see Table 1). Study 2 intentionally focused recruitment efforts on males based on respondent demographics in the first study being ~84% female and to provide a more balanced sample powered for sex DIF analysis. Like study 1, participants completed demographic information, the revised HRP, Perceived Stress Scale (PSS), and the single-response Physical Activity Questionnaire through Qualtrics. The 10-item PSS was used to evaluate the convergent validity evidence of the revised HRP, and a single-response Physical Activity Questionnaire was used to examine validity evidence.

#### 3.7.2. Measures

All measures used in study 1 were utilized in study 2, with the only exception being changes made to the original 23-item HRP (noted in Table 4). Surveys were introduced in the same order as study 1 using the same survey platform to keep study protocols consistent across study 1 and study 2.

### 3.8. Study 2: Health Resilience Profile (HRP-18)—Rasch Calibration

In the second study, the HRP contained 19 items and a 4-point Likert scale that removed the “neither agree nor disagree” (3) response and asked participants to rate items from “strongly disagree” (1) to “strongly agree” (4). In the final version of the HRP, items 6, 7, 17, and 18 are reverse scored (1 = 4, 2 = 3, 3 = 2, 4 = 1), and a total score is calculated by summing all items. Scores on the final version of the HRP (after results from study 2) range from 18–72, with higher scores indicating higher health resilience.

#### 3.8.1. Model Data Fit

Overall, the data fit the Rasch model well (see Table 5). Except for one item (item 17), all infit and outfit statistics were within the acceptable range of 0.5 to 1.5. One item was flagged due to high outfit statistics: Item 17 (“The actions I take daily impact my health”) with an outfit value of 3.60. This item was eliminated from the final estimation, leaving 18 total items in the HRP. The unidimensionality of the revised scale was also satisfied (an eigenvalue on the first contrast was 2.1), and the items of the revised scale were locally independent of each other (<0.7).

#### 3.8.2. Rating Scale Function

Regular observation distribution was found. The average logit measures advanced as category increased and the outfit statistics fell within the desired range of <2.0. The category thresholds (i.e., boundaries between rating categories) were arranged in sequence (see Panel ‘B’ in Figure 1), which was an improvement from Study 1. Overall, the 4-category rating scale functioned well (see Table 2).

#### 3.8.3. Item-Person Map

Panel ‘B’ of Figure 2 depicts the item-person map for study 2. The map shows that the distribution of the items was improved and well-targeted to the participant’s resilience level.

#### 3.8.4. Item Difficulty and Person Ability

The resilience item difficulty estimates ranged from −3.57 to 3.40 logits. A higher logit score indicated a more difficult resilient item. The most difficult resilience item was question 18 (“I get discouraged when I’m not making progress toward my health goal(s)”) with a 3.40 logit (SE = 0.08). The least difficult resilient item was question 13 (“I feel a sense of pride when I reach a health goal”) with a −3.57 logit (SE = 0.11) (see Table 5). The item separation index was 15.87, indicating that the HRP items were well distributed across the measurement scale. The separation reliability was 1.00, indicating a high degree of confidence in replicating the position of the items within measurement error for another sample.

The individual level of resilience was estimated by logit, where a higher logit represented a higher resilience level. The average resilience level was 1.47 (SD = 1.69). The individual level of the resilience estimates ranged from −2.20 to 8.77 logits. Person separation was 2.83, denoting that person’s ability are well separated along the measurement continuum (around three different levels of person level). Person separation reliability was 0.89, signifying an acceptable degree of confidence in replicating the placement of persons within a measurement error.

#### 3.8.5. Differential Item Functioning (DIF)

According to the DIF analysis, there were no items flagged by all subgroups (i.e., age, gender, alcohol use, obesity status, and sex), reflecting that the items function properly regardless of the subgroups. In other words, the items are equivalent in evaluating the resilience level of all people.

#### 3.8.6. Validity Evidence

To assess the evidence of convergent validity of the HRP, the resilience level and perceived stress level (PSS) were compared. There was a moderate negative relationship (*r* = −0.558, *p* < 0.001) between the two scores, which supported the convergent validity evidence of the resilience scale.

To examine a known-group difference validity, the resilience level was compared to physical activity level (low vs. moderate to vigorous). A low physical activity level group (*M* logits = 1.24) has a lower resilience level than a moderate to vigorous level physical activity group (*M* logits = 2.00), which supported the known-group difference validity evidence [40]. An independent *t*-test indicated a significant difference between levels of physical activity, *t* = 4.75, *p* < 0.001).

### 3.9. HRP Revisions Following Study 2

Revisions to the HRP after the second study were minimal. The only change was to remove Item 17 due to infit/outfit statistics outside the acceptable range.

## 4. Discussion

Overall, the results from the two studies support using the 18-item HRP as a measure of resilience among adults enrolled in a health coaching program. Study 1 aimed to develop and examine the psychometric properties of the initial 23-item Health Resilience Profile (HRP) in a sample of adults enrolled in a health coaching program. Findings from study 1 informed modifications to the HRP, and the purpose of study 2 was to use Rasch analysis to evaluate the performance of the revised HRP. Results from study 2 indicated that the 18-item HRP performs well in adults enrolled in a health coaching program targeting physical activity and dietary behavior change. Findings from these studies offer a promising future direction for research to explore how resilience can be measured and integrated into health coaching programs when individuals face challenges and adversity specific to behavior change and goal attainment.

The final version of the HRP contained 18 items with acceptable model-data fit, with only one item demonstrating poor category function (item 17 was removed to yield 18 items). After removing the “neither agree nor disagree” response option based on results from the first study indicating unacceptable response option performance, the 18-item HRP rating scale functioned well with a 4-point Likert (strongly disagree, disagree, agree, strongly agree). Unlike the ceiling effect observed in the CD-RISC with a sample of adults enrolled in a health coaching program, the HRP appears to be an improved measure of resilience with an item separation index of 15.87, indicating that the HRP items were well distributed across participant ability levels [26]. Difficulty estimates ranged from −3.57 to 3.40 logits, which suggests the HRP was not too easy or too difficult for this sample. Furthermore, the 18-item HRP did not function differently because of demographics such as age (young adults, middle-aged adults, or older adults), sex (male or female), alcohol use (yes or no), or obesity status (person with obesity or person without obesity). Validity evidence in the current study showed an inverse relationship between HRP and perceived stress, and individuals with higher physical activity reported higher resilience levels. The findings are consistent with previous findings on the relationships between resilience, stress, and physical activity [41,42]. This is an important consideration to ensure that the inventory is unbiased toward a subgroup based on other factors outside of individual resilience level.

A review of resilience measures asserts that there is no current ‘gold standard’ across 15 measures of resilience [9]. Among resilience measures, the most frequently used are the Connor–Davidson Resilience Scale (CD-RISC), the 10-item CD-RISC, the Resilience Scale for Adults (RSA), and the Brief Resilience Scale (BRS) [43]. The CD-RISC has better psychometric properties than other resilience measures and is one of the most widely used resilience inventories [9,43,44]; however, no instrument has been calibrated for use in a population of adults in a health coaching program. In a psychometric evaluation of the CD-RISC in a sample of adults enrolled in a health coaching program, the instrument performed poorly within that context [26]. Compared to the evaluation of the 25-item CD-RISC in this sample, the current study yielded an improved measure of resilience for adults enrolled in a health coaching program [26]. This was accomplished in two phases. First, we removed nine total misfit items (four items removed from the 25-item CD-RISC, four items removed from the first version of the HRP, and one item removed from the final version of the HRP). Next, we developed and added three items to the HRP that were believed to be necessary to resilience in the context of behavior change, changed the Likert scale from a 5-point to a 4-point scale by removing “Neither agree nor disagree” option, and revised all item wording to fit the context of goal-setting and behavior modification in a health coaching setting. The findings from the present study suggest that the 18-item HRP could be a welcome addition to measures of resilience within an understudied population (adults enrolled in health coaching). The 18-item HRP allows for resilience to be measured and studied beyond samples that past research has primarily focused on, such as caregivers and people who experience medical or natural disasters [43,45].

The present study offers a psychometrically sound instrument to measure resilience in health behavior interventions and the context of adults in health coaching programs. With the addition of improved measurement in this sample, the next steps are to examine changes in resilience over time. Resilience is a skill that can be developed [46]. Several interventions have been proposed to build resilience by enhancing psychosocial factors and behaviors [47]. Ferreira (2021) conducted a systematic review containing 33 randomized controlled trials to determine the success of resilience interventions in improving resilience in adults [48]. Findings suggest that despite methodological shortcomings (including subpar resilience measures), resilience interventions significantly increased resilience and mental health improvements [48]. Resilience interventions utilize a variety of theoretical frames to build programs to change resilience [48]. These paradigms include, but are not limited to: cognitive behavioral therapy, positive psychology, mindfulness meditation, acceptance and commitment therapy, attention and interpretation therapy, and mind–body training to increase resilience levels [48]. Resilience interventions focus on groups susceptible to stress and adversity: military veterans, active military, healthcare workers, caregivers, and individuals diagnosed with cancer [49,50,51,52,53]. Health coaching programs have only recently begun focusing on resilience as a target behavior/skill and a key outcome of interest. Significant increases in youth resilience were reported in a single group intervention study where sixth graders participated in 1:1 health coaching sessions focused on resilience (up to 6 appointments over eight weeks) [54]. This work provides preliminary support for health coaching as a viable strategy to improve resilience in youth [54]. It is unclear if adults enrolled in health coaching would experience similar improvements in resilience.

This study is not without limitations. First, both study 1 and study 2 were conducted several months after the onset of the SARS-CoV-2 pandemic. Since cultural and temporal contexts influence resilience, it could be that the HRP was shown to be an improved measure of resilience compared to the 25-item CD-RISC (26; conducted prior to the SARS-CoV-2 pandemic in a similar sample), in part, because of collective trauma [55]. Some scholars contend that a person’s culture and cultural values should be placed at the core of resilience, the COVID-19 pandemic likely influenced participant responses about resilience collectively [55,56]. Future work should include an inventory that measures the impact of COVID-19 on a person’s employment, income, childcare, and changes in responsibility for child(ren)’s schooling to understand health-related resilience and COVID-19 impact better. Another limitation is the generalizability of the current study findings. While we oversampled men in study 2 to assess differential item functioning between sexes, our sample was still 62.1% women. Further, most participants in the current study reported completing a college education or more and had a household income of $75,000 or greater. This suggests that our sample does not reflect individuals from lower socioeconomic backgrounds and less educated individuals. This is particularly significant for resilience research since individuals with lower socioeconomic status are more susceptible to acute and chronic exposure to stress than others [57]. It is essential to note that the overarching aim of this work was to develop a resilience tool that functioned well in a sample of adults enrolled in a health coaching program. This study recruited an adequate sample size representative of adults attempting behavior change enrolled in health coaching programs. It is recommended that future use of the HRP be utilized in similar samples and contexts related to health coaching.

The current study has several notable strengths. Presenting these combined data from a two-part study allows readers to observe and understand the complete process to create, identify issues, and then optimize an instrument. First, the development of the 23-item and 18-item HRP was guided by previous work with similar samples. The initial 23-item HRP was based on Rasch analysis results of the CD-RISC in a sample of adults enrolled in a health coaching program [26]. Additionally, study 1 findings supported the changes made to the HRP reflected in the second study. In the initial development of the 23 items, two co-authors with a combined 30 years of experience working in health coaching programs reviewed and revised items to be tailored to adults enrolled in health coaching programs (use of phrases such as “health goals” and applying resilience to the context of goal attainment). Items were then reviewed by other health coaching program personnel (registered dietitians and health coaches) to determine if items were worded appropriately for individuals enrolled in health coaching programs. The phrasing of the HRP was intentionally specific (“health goals”) yet also vague enough to ensure its application to a wide range of health behavior settings. It is encouraged that future use of the 18-item HRP add a statement to the instructions that contextualize what is meant by the phrase “health goal” in their specific setting. Doing so will tailor this measure to different health behavior change settings and allow respondents to select relevant and meaningful responses. For example, a practitioner working with someone on sleep hygiene can define a goal in this setting to help participants answer in a way that is both relevant and accommodating to their specific health interests. For other practitioners and researchers using the scale to examine resilience more generally, it is advised to keep the HRP instructions the same as that presented in Appendix A, Table A1. Another strength of the current study includes the potential to translate this work into practice. The study team consists of an interdisciplinary collaboration between industry and academia. It is expected that the HRP will be integrated into the Profile by Sanford health coaching program to understand how resilience may change throughout a person’s membership/duration in the program. This integration of the HRP into practice is consistent with the mission of the National Institutes of Health (NIH) to investigate “fundamental knowledge about the nature and behavior of living systems and the application of that knowledge to enhance health, lengthen life, and reduce illness and disability” [58].

Future research should determine the efficacy of the HRP as an outcome measure in programming designed to improve resilience offered as a standalone program or incorporated into a larger health coaching program. For example, integrating a resilience training program into a health coaching program and measuring resilience outcomes using the HRP would allow programs to understand the efficacy of resilience training programs and if the HRP can adequately measure changes in resilience over time. Additionally, future research could evaluate the predictive utility of resilience (using the HRP) on program adherence and attrition in adults enrolled in a health coaching program. Improved understanding of how resilience may impact key health outcomes (such as quality of life, adherence to physical activity and dietary change, and attrition/dropout) for adults enrolled in a health coaching program could inform programming efforts to tailor treatments that cultivate resilience at critical times or after significant events that occur in the behavior change process. The current study assessed resilience in a health coaching program focused on nutrition, activity, and lifestyle habits. However, the items were worded so that the HRP could be used in other health and life coaching programs (for example, intuitive eating, mindfulness, personal training, and weight-inclusive/nondieting services). It would be worthwhile to assess the psychometric properties in diverse settings to determine if the 18-item HRP performs well across different programming contexts.

## 5. Conclusions

The present study is aligned with Tsai and Freedland’s (2022) assertion that resilience deserves more in-depth study within the field of health psychology [59]. Resilience is associated with improved mental health and reduced perceived stress [60]. Incorporating resilience into health coaching programs for adults may improve physical and mental health. Studies focused on understanding resilience have increased in response to the SARS-CoV-2 pandemic, with calls for people, communities, and societies to “build resilience” [61,62,63,64]. As the scientific study of resilience increases in response to the COVID-19 pandemic and beyond, it is critical to evaluate how various resilience measures perform in different samples and contexts/settings. This study contributes to the resilience and public health literature by offering a psychometrically robust measure (18-item HRP) to study resilience in the context of adults attempting physical activity and dietary behavior changes in a health coaching program where goal-setting is encouraged. The 18-item HRP could be used in public health programs and health coaching programs in two ways: (1) Encourage programs to incorporate resilience-building exercises or activities into programming and use the HRP to evaluate resilience outcomes and (2) improve the tailoring of health coaching services based upon an individual’s resilience level. Both pathways may reduce attrition and adherence issues that commonly occur in behavior change and health coaching programs.

## Figures and Tables

**Figure 1 nutrients-15-00807-f001:**
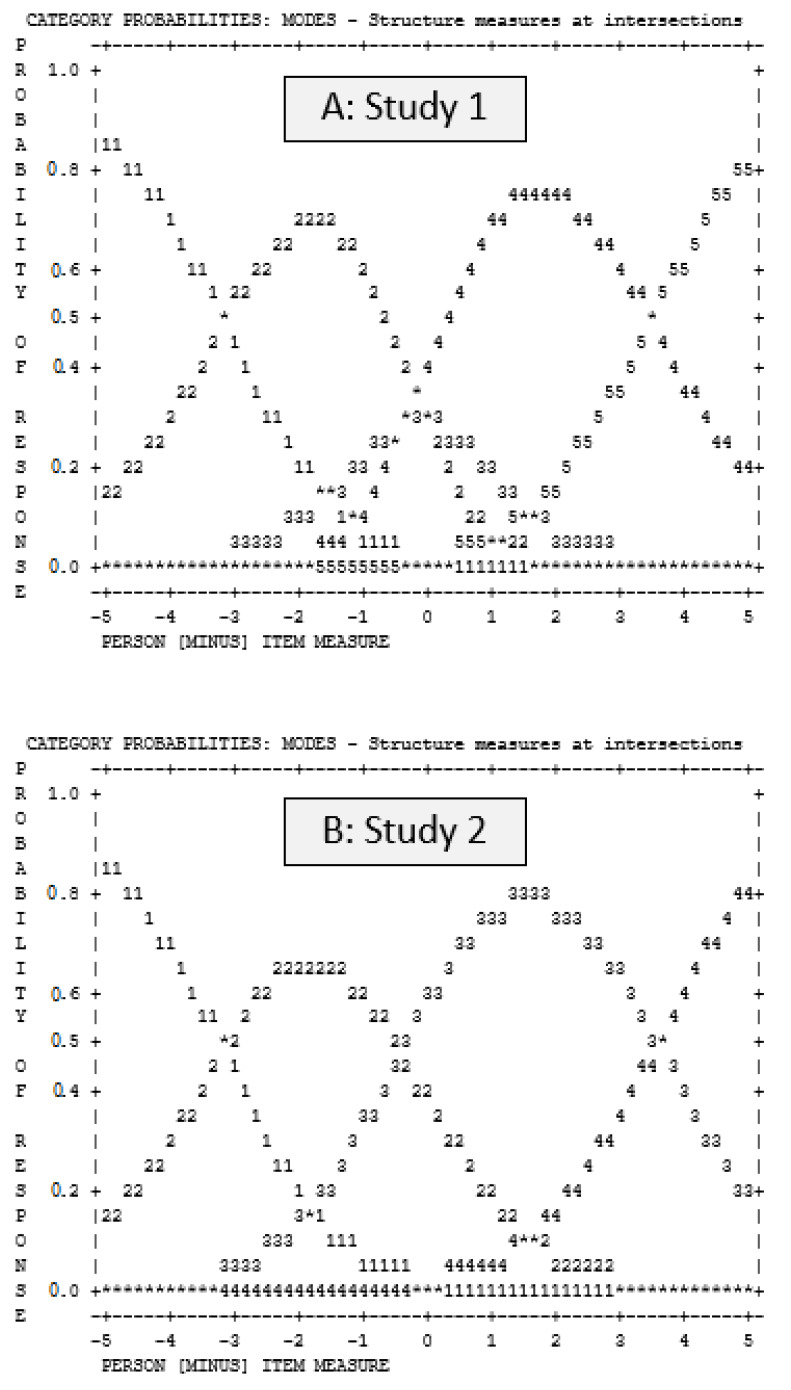
HRP category probabilities from study 1 (panel ‘A’) and study 2 (panel ‘B’).

**Figure 2 nutrients-15-00807-f002:**
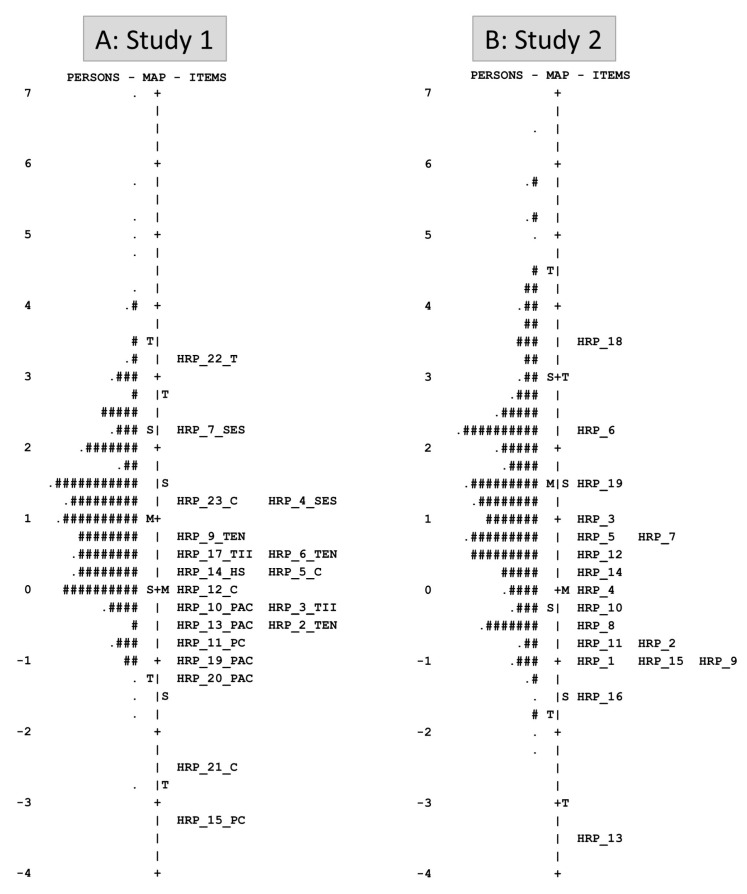
HRP item–person map from study 1 (panel ‘A’) and study 2 (panel ‘B’).

**Table 1 nutrients-15-00807-t001:** Participant characteristics for study 1 and study 2.

Characteristics	Study 1 *n* = 427		Study 2 *n* = 493	
	*M* ± SD	%	*M* ± SD	%
Age (years)	44.46 ± 11.93	12.2% (52) missing	49.76 ± 12.53	
Young adults (18–35 years)	22.2%			15.6%
Middle-aged adults (36–55 years)	45.7%			48.7%
Older adults (≥56 years)	19.9%			35.7%
Sex (%)		12.2% (52) missing		
Male	4%			37.7%
Female	83.8%			62.1%
Prefer not to answer				0.2%
BMI (kg/m^2^)				
Non-obese (38.3%)	26.43 ± 2.41		26.53 ± 2.52	
Obese (61.7%)	37.11 ± 6.28		36.46 ± 5.34	
Alcohol usage		12.4% (53) missing		
Yes	37.9%			44.2%
No	49.6%			55.8%

BMI = Body Mass Index.

**Table 2 nutrients-15-00807-t002:** Summary of study 1 and study 2 HRP rating scale function.

Category Score	Counts Used	Average Measure	Outfit MNSQ	Category Thresholds
Study 1				
1	265	−2.06	1.26	None
2	1462	−0.81	1.02	−3.16
3 *	1154	0.26	0.86	0.01 *
4 *	3940	1.53	1.01	−0.34 *
5 *	1273	3.35	1.02	3.50 *
Study 2				
1	331	−1.85	1.64	None
2	1907	−0.68	0.81	−3.18
3	4760	1.54	0.98	−0.41
4	1822	3.91	1.01	3.59

Average measure: a mean of logit measures in category; MNSQ: mean square residuals. * = indicates category thresholds not arranged in sequence.

**Table 3 nutrients-15-00807-t003:** Item summary (*n* = 19) of Rasch calibration in HRP-23 scale from study 1.

Item	Calibration Logits	SE Logits	Infit MNSQ	Outfit MNSQ
Q22. I get discouraged when I’m not making progress toward my health goal(s)	3.15	0.08	1.33	1.29
Q7. Unpleasant feelings like sadness, anger, fear, and boredom get in the way of my health goal(s)	2.23	0.07	1.26	1.31
Q4. I stay focused on my health goal(s) even when feeling overwhelmed	1.19	0.06	0.97	0.99
Q23. Outside factors keep me from pursuing my health goal(s)	1.18	0.06	1.05	1.10
Q9. I give up on my health goal(s) when the challenges of everyday life get in the way	0.84	0.06	0.84	0.89
Q6. I work to reach my health goal(s) no matter what roadblocks or challenges I come across in the process	0.56	0.07	0.94	0.96
Q17. When unexpected challenges occur while pursuing my health goal(s), I know how to handle them	0.47	0.07	0.72	0.73
Q14. I give my best effort to reach my health goal(s), no matter what happens	0.27	0.07	0.94	0.98
Q5. I feel in control of my health goal(s)	0.16	0.07	0.88	0.89
Q12. When I feel overwhelmed, I know how to get the help I need	−0.12	0.07	1.02	1.04
Q3. I can make difficult decisions to reach my health goal(s) when necessary	−0.29	0.07	0.82	0.84
Q10. I tend to bounce back after experiencing setbacks	−0.34	0.07	1.07	1.13
Q13. I have confidence from past successes that impacts how I handle challenges and difficulties	−0.38	0.07	0.93	0.99
Q2. I enjoy setting health goal(s) that challenge me	−0.39	0.07	1.21	1.32
Q11. I believe I can achieve my health goal(s), even if there are obstacles	−0.73	0.08	0.83	0.77
Q19. I am able to apply what I’ve learned from past failures	−0.92	0.08	0.79	0.79
Q20. I believe past failures contribute to my growth	−1.17	0.19	0.98	1.03
Q21. The actions I take daily impact my health	−2.54	0.10	1.11	1.10
Q15. I feel a sense of pride when I reach a health goal	−3.18	0.11	1.13	1.08

SE = standard errors; MNSQ = mean square residuals.

**Table 4 nutrients-15-00807-t004:** Pilot test item revisions and deletion with justification.

Item	Modification	Rationale
16. I am my biggest critic when it comes to my health goal(s)	Item removed	Item demonstrated poor infit/outfit statistics in study 1
1. I am not easily discouraged by failure	Item removed	Item demonstrated poor infit/outfit statistics in study 1
8. My health goal(s) are connected to my purpose in life	Item removed	Item demonstrated poor infit/outfit statistics in study 1
18. I have people I can rely on to support me when I need it	Item removed	Item demonstrated poor infit/outfit statistics in study 1
5-point Likert response	Changed from a 5-point to a 4-point response	The 5-point response did not perform well. As such, the “neither agree nor disagree” response was removed in study 2 to create a 4-point Likert scale.

**Table 5 nutrients-15-00807-t005:** Item difficulty summary of Rasch calibration in the 18-item HRP.

Item #	Item	Calibration Logits (SE)	Infit MNSQ	Outfit MNSQ
18	I get discouraged when I’m not making progress toward my health goal(s)	3.40 (0.08)	1.26	1.31
6	Unpleasant feelings like sadness, anger, fear, and boredom get in the way of my health goal(s)	2.32 (0.08)	1.36	1.41
19	Outside factors keep me from pursuing my health goal(s)	1.40 (0.08)	1.23	1.30
3	I stay focused on my health goal(s) even when feeling overwhelmed	1.12 (0.08)	0.84	0.84
7	I give up on my health goal(s) when the challenges of everyday life get in the way	0.82 (0.09)	0.99	1.02
5	I work to reach my health goal(s) no matter what roadblocks or challenges I come across in the process	0.74 (0.09)	0.90	0.89
12	I give my best effort to reach my health goal(s), no matter what happens	0.38 (0.09)	0.77	0.74
14	When unexpected challenges occur while pursuing my health goal(s), I know how to handle them	0.25 (0.09)	0.59	0.59
4	I feel in control of my health goal(s)	−0.04 (0.09)	0.90	0.94
10	When I feel overwhelmed, I know how to get the help I need	−0.31 (0.09)	1.07	1.03
8	I tend to bounce back after experiencing setbacks	−0.53 (0.09)	1.05	1.07
11	I have confidence from past successes that impacts how I handle challenges and difficulties	−0.75 (0.09)	0.86	0.80
2	I can make difficult decisions to reach my health goal(s) when necessary	−0.85 (0.09)	0.84	0.80
9	I believe I can achieve my health goal(s), even if there are obstacles	−0.93 (0.09)	0.81	0.77
15	I am able to apply what I’ve learned from past failures	−0.95 (0.09)	0.75	0.74
1	I enjoy setting health goal(s) that challenge me	−0.98 (0.09)	1.45	1.47
16	I believe past failures contribute to my growth	−1.50 (0.09)	0.91	1.02
13	I feel a sense of pride when I reach a health goal	−3.57 (0.09)	1.29	1.39

## Data Availability

Data available upon request to corresponding author.

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
