# Peer review of "Development and Rasch Analysis of the 18-Item Health Resilience Profile (HRP)"

_nutrients, 2023, doi:10.3390/nu15040807_

Round 1
Reviewer 1 Report
Thank you for the opportunity to review this manuscript regarding the development of the 18-item Health Resilience Profile (HRP). Overall, I think the authors have made a compelling argument for the importance of work and have offered meaningful suggestions for future research regarding resilience and use of their measure. To further strengthen this work, I offer a few comments that I would like the authors to consider. I hope the authors find my review of their manuscript helpful.
ABSTRACT
1. The opening sentence of the abstract should be about the shortcoming of existing resilience instruments rather than resilience potentially improving adherence to a health coaching program.
INTRODUCTION
2. The opening introductory sentence (lines 27-30) is a bit hard to follow and would benefit from minor restructuring and revision.
3. The opening paragraph addresses how attrition is a challenge in behavior change research. The second paragraph notes how resilience is important for managing stress and bouncing back from adversity. The authors need to talk about reasons for attrition, which may be for reasons other than stress and adversity. Resilience may be important if these are the reasons for attrition, but data and citations should be provided that speak to this point.
4. Lines 52-56: The authors note that navigating social settings, lack of knowledge, monotony, and boredom are barriers in health coaching programs that address diet and nutrition. The authors should also speak to other common barriers such as time constraints, competing obligations, and access-related issues. The authors should also talk about how resilience might be cultivated to address the most common barriers to behavior change.
5. Paragraphs 4, 5, and 6 are somewhat at odds with one another. For example, paragraph 4 notes that “…coaching can support increased resilience in various populations and contexts,…” Paragraph 5 then goes on to imply that current resilience measures are lacking/limited, and paragraph 6 addresses various issues with resilience measures. If there are issues with current resilience measures, then how strong is the research showing that coaching can support increased research (paragraph 4)?
6. It may be more intuitive to talk about the current resilience measures and their limitations (lines 68-84) before talking about a Rasch model and how such a model is preferred over traditional methods. (lines 57-67)
METHODS
7. Lines 112-114. Is there any evidence to support that an Infit value of 0.5 and an Outfit value greater than 1.5 reflect poor fit? A citation would be helpful here.
8. Please provide more information about the physical activity measure. For example, what was the question and what were the response options? Can evidence by provided to show that this question is acceptable for purposes of assessing physical activity?
RESULTS
9. I commend the authors on the level of detail provided in the results section. While there’s a lot to unpack, all of the information is important to present.
DISCUSSION
10. The authors have provided thoughtful and appropriate discussion points relevant to their analyses. Although the focus of the manuscript is not about interventions to enhance resilience, I think it would be helpful if the authors could speak to how resilience interventions promote resilience. The authors note in lines 438-439 that “Several interventions have been proposed to build resilience by enhancing psychological factors and behaviors.” How is this actually done?
11. The authors note that a one limitation is that findings are limited to adults enrolled in a health coaching program. However, it seems as if a major point of the authors work is that this scale is designed for adults in health coaching programs. Therefore, I wouldn’t consider that only having participants in health coaching programs is an actual limitation. They authors speak to this point by noting the aim of their work, so I would just recommend deleting the text that notes this as a limitation.
CONCLUSIONS
12. The concluding remarks are appropriate for the work that was performed.
Author Response
Dear reviewer,
Thank you for your review of our manuscript titled: “Development and Rasch Analysis of the 18-item Health Resilience Profile (HRP).” We have made additional revisions based upon your recommendations. Please see detailed responses to your comments below. We are hopeful these revisions will result in a favorable review, and most importantly, a stronger manuscript.
Thanks again for your review of our paper.
Reviewer 1
ABSTRACT
- The opening sentence of the abstract should be about the shortcoming of existing resilience instruments rather than resilience potentially improving adherence to a health coaching program.
Thank you for your feedback on the abstract introduction. We have modified this to read as follows:
Existing resilience measures have psychometric shortcomings, and there is no current gold-standard resilience measure. Previous work indicates adults enrolled in a health coaching program may benefit from a resilience measure that is tailored and contextualized to this sample.
PAGE 1, LINES 11-13
INTRODUCTION
- The opening introductory sentence (lines 27-30) is a bit hard to follow and would benefit from minor restructuring and revision.
We have modified this section to improve clarity. Please see page 2:
The optimal dose of health coaching for physical activity, dietary, stress management, and smoking behavior change is 30 or more sessions over the course of 6 to 12 months [1].
PAGE 2, LINES 29-31
- The opening paragraph addresses how attrition is a challenge in behavior change research. The second paragraph notes how resilience is important for managing stress and bouncing back from adversity. The authors need to talk about reasons for attrition, which may be for reasons other than stress and adversity. Resilience may be important if these are the reasons for attrition, but data and citations should be provided that speak to this point.
We agree that we should better tie in the first paragraph to the following paragraph. To do this, we elaborated on the previous citation [#5] to include that people in this study reported various behavioral, demographic, and social barriers to participation and reasons for drop out. This ties in with the concept of resilience since the operational definition of resilience in our paper is the ability to bounce back in the face of such barriers. Please see:
People enrolled in lifestyle management program report a variety of social, demographic, and behavioral barriers for participation and reasons for drop out [5].
PAGE 1, LINES 34-35 - Lines 52-56: The authors note that navigating social settings, lack of knowledge, monotony, and boredom are barriers in health coaching programs that address diet and nutrition. The authors should also speak to other common barriers such as time constraints, competing obligations, and access-related issues. The authors should also talk about how resilience might be cultivated to address the most common barriers to behavior change.
Thank you for this comment. A wide range of potential barriers have been reported in various populations. The intent is not to provide an exhaustive list of these barriers but to provide a relevant example. Further, the citation provided (Kleine 2019) was a previous qualitative study in participants from the same health coaching program that we recruited and conducted this HRP study. Kleine et al used focus groups to identify barriers and identified the list provided as the top themes. As such, we feel these particular reasons are relevant to the present study since it comes from participants enrolled in the same program.
While we agree it would be helpful to outline how resilience could be cultivated within this sample, the purpose of this paper is to first determine an adequate measure of resilience for this sample. We believe it is outside of the scope of this introduction to detail how to increase resilience specific to the noted barriers of a health coaching program. However, to emphasize the second part of this reviewer comment, we added the following statement to better conclude why resilience is necessary:
Increasing resilience may be a worthwhile pursuit for health coaching programs to help individuals overcome and persist in light of these barriers.
- Paragraphs 4, 5, and 6 are somewhat at odds with one another. For example, paragraph 4 notes that “…coaching can support increased resilience in various populations and contexts,…” Paragraph 5 then goes on to imply that current resilience measures are lacking/limited, and paragraph 6 addresses various issues with resilience measures. If there are issues with current resilience measures, then how strong is the research showing that coaching can support increased research (paragraph 4)?
Great point, thank you for providing this feedback. We have added the following sentence in paragraph 4 to help alleviate some of the conflicting evidence here: However, findings from resilience interventions should be interpreted with caution since there no “gold standard” resilience measures [9].
PAGE 2, LINES 67-69
- It may be more intuitive to talk about the current resilience measures and their limitations (lines 68-84) before talking about a Rasch model and how such a model is preferred over traditional methods. (lines 57-67)
We appreciate this feedback on order and clarity of the introduction. We believe it is better to first introduce why Rasch analysis is ideal in this scenario and then to recap the limitations of existing resilience measures. To improve this transition, we have made slight modification in the text so readers can view the discussion of existing instruments and limitations through a psychometric lens following the rasch measurement paragraph.
The outline of our introduction is as follows:
A. Introduce health coaching-specific issues
- Introduce and define resilience and that it is a skill that can be cultivated
C. Why measurement matters and how Rasch analysis is preferred
D. Psychometric limitations of existing resilience measures
E. Present study aims/purpose
METHODS
- Lines 112-114. Is there any evidence to support that an Infit value of 0.5 and an Outfit value greater than 1.5 reflect poor fit? A citation would be helpful here.
We included the appropriate citation to support this statement (Linacre, 2002; citation #29).
PAGE 3, LINE 132
- Please provide more information about the physical activity measure. For example, what was the question and what were the response options? Can evidence by provided to show that this question is acceptable for purposes of assessing physical activity?
We have addressed this comment by providing more information:
This single-response Physical Activity Questionnaire invites participants to select one of eight possible responses that best matches their current level of physical activity. Participants who respond with the fifth response or higher are meeting or exceed physical activity guideline recommendations [33].
PAGE 5, LINES 186-189
RESULTS
- I commend the authors on the level of detail provided in the results section. While there’s a lot to unpack, all of the information is important to present.
We appreciate your review and comment on the results section.
DISCUSSION
- The authors have provided thoughtful and appropriate discussion points relevant to their analyses. Although the focus of the manuscript is not about interventions to enhance resilience, I think it would be helpful if the authors could speak to how resilience interventions promote resilience. The authors note in lines 438-439 that “Several interventions have been proposed to build resilience by enhancing psychological factors and behaviors.” How is this actually done?
Thank you for pointing out this oversight. We also believe this should be added to the discussion to help readers better understand various applications of this work. Please see this revision:
Resilience interventions utilize a variety of theoretical frames to build programs to change resilience [48]. These paradigms include, but are not limited to: cognitive behavioral therapy, positive psychology, mindfulness meditation, acceptance and commitment therapy, attention and interpretation therapy, and mind-body training to increase resilience levels [48].
PAGE 14, LINES 464-468
- The authors note that a one limitation is that findings are limited to adults enrolled in a health coaching program. However, it seems as if a major point of the authors work is that this scale is designed for adults in health coaching programs. Therefore, I wouldn’t consider that only having participants in health coaching programs is an actual limitation. They authors speak to this point by noting the aim of their work, so I would just recommend deleting the text that notes this as a limitation.
Thank you for this feedback. Our purpose in including this in the limitations section was to communicate caution in applying this measure to other samples who are very different in certain sociodemographic characteristics. We revised this section to read as follows:
Another limitation is that the generalizability of the current study findings. While we oversampled men in study 2 to assess differential item functioning between sexes, our sample was still 62.1% women. Further, most participants in the current study reported completing a college education or more and had a household income of $75,000 or greater. This suggests that our sample does not reflect individuals from lower socioeconomic backgrounds and less educated individuals. This is particularly significant for resilience research since individuals with lower socioeconomic status are more susceptible to acute and chronic exposure to stress than others [57]. It is essential to note that the overarching aim of this work was to develop a resilience tool that functioned well in a sample of adults enrolled in a health coaching program. This study recruited an adequate sample size representative of adults attempting behavior change enrolled in health coaching pro-grams. It is recommended that future use of the HRP be utilized in similar samples and contexts related to health coaching.
PAGE 15, LINES 489-501
CONCLUSIONS
- The concluding remarks are appropriate for the work that was performed.
Thank you for your review. We appreciate your time and sharing your expertise with us. We believe our manuscript is better for it and hope that you are pleased with our revisions.
Reviewer 2 Report
A nice description of another scale/analysis/intepretation on matter of Health resilience Profile.
It is a good effort and the paper has some interest for the readers of the journal.
Introduction sets the scene well and provides a very good rationale on this new instrument and how this adds on on the previous and existent aspects on the current literature.
Methods are ok and extensively described. No comments here.
REsults are very extensive and give some validity of this new instrument. For this reviewer it looks ok and it seems to do the job, but still at some point the Editor has to keep in mind that the present editor has some limited experience in this advanced statistics part.
Discussion section is truly excellent with good and exhaustive comparisons on the new and old instruments in this matter. They really did a good job here.
Conclusions are balanced and ok.
I think this manuscript can be accepted as it, still considering the statistical limitations of this reviewer, mentioned above.
Author Response
Dear reviewer,
Thank you for your review of our manuscript titled: “Development and Rasch Analysis of the 18-item Health Resilience Profile (HRP).” Please see responses to your comments below.
Thanks again for your review of our paper.
Reviewer 2
A nice description of another scale/analysis/intepretation on matter of Health resilience Profile.
It is a good effort and the paper has some interest for the readers of the journal.
Thank you for your time and effort reviewing our paper.
Introduction sets the scene well and provides a very good rationale on this new instrument and how this adds on on the previous and existent aspects on the current literature.
We also believe the introduction does a nice job of explaining why we conducted this two-part study. We appreciate your comment.
Methods are ok and extensively described. No comments here.
Thank you.
REsults are very extensive and give some validity of this new instrument. For this reviewer it looks ok and it seems to do the job, but still at some point the Editor has to keep in mind that the present editor has some limited experience in this advanced statistics part.
Thank you for sharing.
Discussion section is truly excellent with good and exhaustive comparisons on the new and old instruments in this matter. They really did a good job here.
Thank you for your feedback on our work. We tried to provide a balanced view of our work and how it can be applied in the future.
Conclusions are balanced and ok.
Thank you.
I think this manuscript can be accepted as it, still considering the statistical limitations of this reviewer, mentioned above.
Thank you for your time and review. We appreciate your willingness to review our manuscript and provide feedback.
